# Direct Conversion of CO$_2$ into Dimethyl Ether over Al$_2$O$_3$/Cu/ZnO Catalysts Prepared by Sequential Precipitation

**Cheonwoo Jeong [1,2,†], Jinsung Kim [2,†], Ji-Hyeon Kim [3,4] 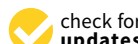, Sunghoon Lee [2] , Jong Wook Bae [3] and Young-Woong Suh [2,5,*]**

[1] Gwangyang Research Group, Research Institute of Industrial Science & Technology, Gwangyang 57801, Korea; cjeong@rist.re.kr
[2] Department of Chemical Engineering, Hanyang University, Seoul 04763, Korea; jinsung0716@hanmail.net (J.K.); goldsepud@hanyang.ac.kr (S.L.)
[3] School of Chemical Engineering, Sungkyunkwan University, Suwon 16419, Korea; kimjyun@iae.re.kr (J.-H.K.); finejw@skku.edu (J.W.B.)
[4] Plant Engineering Division, Institute for Advanced Engineering, Yongin 17180, Korea
[5] Research Institute of Industrial Science, Hanyang University, Seoul 04763, Korea
[*] Correspondence: ywsuh@hanyang.ac.kr; Tel.: +82-2-2220-2329
[†] These authors equally contributed to this work.

**Abstract:** Bifunctional Al$_2$O$_3$/Cu/ZnO catalysts with Al composition of between 30 mol% and 80 mol% were prepared by sequential precipitation (SP) for the conversion of CO$_2$ into dimethyl ether (DME). In the SP synthesis, the concentration of a precipitation agent managed to be high enough to induce the complete precipitation of Al$^{3+}$. The prepared precipitates were composed of zincian malachite and amorphous AlO(OH). Furthermore, the calcined mixed metal oxide materials of 60% and 80% Al exhibited a higher acidity than commercial Al$_2$O$_3$ and the H$_2$-reduced catalysts showed the similar Cu dispersion of 6%–7% at all Cu loadings. In the activity test at 573 K and 50 bar, the SP-derived catalyst of 80% Al (SP-80) displayed the best performance corresponding to CO$_2$ conversion of 25% and DME selectivity of 75% that are close to equilibrium values. In order to overcome the thermodynamic limitation, a dual-bed catalyst system was made up of SP-80 in the first layer and zeolite ferrierite in the next. This approach enabled DME selectivity to be enhanced to 90% while CO$_2$ conversion increased a little. Consequently, the studied catalyst system based on the SP-derived catalysts can contribute greatly to selective DME production from CO$_2$.

**Keywords:** CO$_2$ conversion; Dimethyl ether; Sequential precipitation; Bifunctional catalyst

## 1. Introduction

To limit the detrimental impacts of climate change caused by the rise in global CO$_2$ concentration, a variety of methods and technologies have been explored worldwide to remove CO$_2$ from the atmosphere and from the flue gas, followed by recycling the CO$_2$ for utilization and securing safe and sustainable storage options, which is the general concept of carbon capture, utilization, and storage (CCUS). Among them, converting CO$_2$ into useful chemicals has been attracting much attention in recent years even if economic issue is still under debate. A well-known chemical is methanol produced by CO$_2$ hydrogenation, which was first commercialized in Iceland [1]. However, the thermodynamic equilibrium limits the methanol synthesis process to a low conversion, thus recycling the outlet stream in order to approach a desired conversion value. Another option to circumvent this limitation is coupling methanol synthesis with methanol-consumed reactions in series. Among a

number of candidates, the most intensively studied reaction pathway is direct conversion of $CO_2$ into dimethyl ether (DME) because methanol intermediate is simply changed into DME via intermolecular dehydration over acid catalysts, and DME can be used as a fuel replacement and converted into valuable chemicals including olefin and gasoline [2].

The usual way to achieve this direct conversion is mixing of Cu/ZnO-based methanol synthesis catalyst with an acid catalyst such as $\gamma$-$Al_2O_3$ and zeolite by physical mixing or precipitation, where the acid catalyst is responsible for methanol dehydration to DME and therefore increases DME selectivity [2–4]. Especially, the elegant catalyst system named "zeolite capsule catalyst" was first suggested by Tsubaki and coworkers, where a microporous zeolite layer was coated onto the outer shell of a Cu/ZnO/$Al_2O_3$ core [5]. We recently explored the similar catalyst concept by adjusting the synthesis protocol of an industrial methanol-synthesis Cu/ZnO/$Al_2O_3$ catalyst instead of adding another acid material [6]. Our strategy is referred to as sequential precipitation (SP), different from homogeneous co-precipitation that produces a well-mixed phase of Cu, ZnO, and $Al_2O_3$ favoring the formation of methanol by CO and/or $CO_2$ hydrogenation [7,8]. The SP-derived $Al_2O_3$/Cu/ZnO catalysts could efficiently convert carbon monoxide into methanol and then into DME, compared to their co-precipitated counterparts, because CO hydrogenation to methanol took place over a greater number of accessible Cu surface atoms and the subsequent dehydration of methanol to DME was facilitated by $Al_2O_3$ acid sites existing at the external particle surface [6]. In other words, $Al_2O_3$ in conventional co-precipitated Cu/ZnO/$Al_2O_3$ is a structural promoter for Cu particles necessary for methanol synthesis [9,10], whereas $Al_2O_3$ in the SP-derived $Al_2O_3$/Cu/ZnO catalysts acts as acid site for the dehydration reaction that is similar to previous studies [11,12]. In addition, the effect of $Al^{3+}$ precipitation onto primitive Cu,Zn particles was studied and the resulting precursor was a mixture of zincian malachite and hydrotalcite leading to more abundant Cu surface sites and improved methanol productivity by $CO_2$ hydrogenation [6]. However, when $Al_2O_3$/Cu/ZnO with Al composition of 30% was tested in the direct synthesis of DME from $CO_2$, DME selectivity was unsatisfactory (shown later). This is attributed absolutely to water molecules formed by $CO_2$ hydrogenation into methanol ($CO_2$ + $3H_2 \leftrightarrow CH_3OH + H_2O$) and by the reverse water-gas shift reaction (rWGS; $CO_2 + H_2 \leftrightarrow CO + H_2O$). Therefore, the produced $H_2O$, of which the amount must be larger than that obtained when CO is only used as carbon source, will slow down the reaction rate of methanol dehydration to DME that is forming water as well. This explains that DME selectivity from $CO_2$ is thermodynamically less than that from CO [3,4,13].

Since such a negative effect of $CO_2$ on DME selectivity is considered to be compromised by a higher acidity of SP-derived $Al_2O_3$/Cu/ZnO, the composition of Al needs to be increased up to 80% in the total metal elements. Thus, the catalysts with 60% and 80% Al were prepared by using a higher concentration of precipitation agent to induce complete precipitation of all metal ions, aside from $Al_2O_3$/Cu/ZnO with 30% and 40% Al. All of the prepared catalysts were tested in direct $CO_2$ conversion to DME. Then, we tried to screen the reaction condition and use $\gamma$-$Al_2O_3$ in the second catalyst bed to improve DME selectivity. To overcome the observed limitation in DME selectivity caused by the thermodynamic equilibrium, we considered an additional reaction that can convert DME because the equilibrium would be shifted to a relatively higher level by the consumption of DME. We attempted to place zeolite ferrierite in the second bed after the first $Al_2O_3$/Cu/ZnO bed, which was successful for higher DME selectivity while methyl acetate was formed by DME carbonylation using CO produced via the rWGS reaction. Consequently, the catalyst systems investigated herein would pave the way for converting $CO_2$ into DME in a selective manner.

## 2. Results and Discussion

### 2.1. Preparation of $Al_2O_3$/Cu/ZnO Catalyst

For $Al_2O_3$/Cu/ZnO catalysts used in this work, a series of Cu,Zn,Al precursors were prepared in such a manner that $Al^{3+}$ was precipitated onto the aged Cu,Zn precipitate, where the nominal Al

composition (= [Al]/{[Cu] + [Zn] + [Al]} × 100%) varied from 30% to 80%. Typically, an aqueous solution of $Cu(NO_3)_2$ and $Zn(NO_3)_2$ ($Cu^{2+}$:$Zn^{2+}$ = 7/3) was added dropwise into $NaHCO_3$ solution at 343 K and aged under vigorous stirring for 60 min, followed by injecting an aqueous $Al(NO_3)_3$ solution and further aging for 30 min. A final precipitate was washed thoroughly and dried overnight, which is hereafter denoted as SP-*x* where *x* indicates Al composition of 30%, 40%, 60%, or 80%. All variables in the synthesis were identical to those described in our previous report [6,14], but the concentration of the starting $NaHCO_3$ solution was changed to 0.16 M due to strong acidity of $Al^{3+}$ solution. For instance, when 0.1 M $NaHCO_3$ solution was used for the preparation of SP-60, a pH value dropped to 4.1 by addition of $Al^{3+}$ solution and was then stabilized around 4.5 at the end of aging (Figure 1a). Since this value is below the minimum pH for complete precipitation [15], the concentrated $NaHCO_3$ solution of 0.16 M was used, thus approaching a pH of ca. 6.6 for SP-60 and SP-80 (Figure 1b).

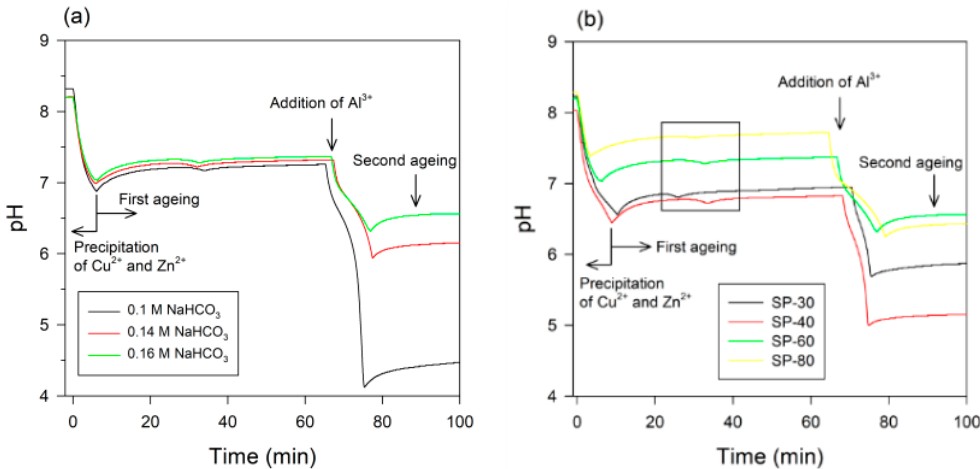

**Figure 1.** pH profiles in the precipitation and aging stages for (**a**) SP-60 precursors obtained when an aqueous $NaHCO_3$ solution of 0.1, 0.14 or 0.16 M was employed, and (**b**) SP-30 and SP-40 precursors prepared by 0.1 M $NaHCO_3$ and SP-60 and SP-80 precursors prepared by 0.16 M $NaHCO_3$.

When the SP-derived precursors were characterized by XRD analysis, the common feature was no detection of the (003) reflection of hydrotalcite phase (Figure 2a), unlike Cu, Zn, and Al precursors prepared by co-precipitation [16]. Most reflections correspond to a phase of $Zn^{2+}$-displaced malachite [zM; $(Cu_{1-x}Zn_x)_2(OH)_2CO_3$]. These results are associated with precipitation of $Al^{3+}$ onto the fully developed CuZn co-precipitate, which is achieved by addition of $Al(NO_3)_3$ solution after pH drop (marked by a rectangle in Figure 1b) indicating the transformation of amorphous precipitate to crystalline zM [17]. On the other hand, the (20 − 1) reflection of zM phase was shifted to lower $2\theta$ angles and became less intense with Al composition increasing, due to lower presence of Cu and Zn. Meanwhile, broad humps centered around $2\theta$ of 14°, 28°, and 38° were distinct for SP-60 and SP-80. A similarity was found for the precipitate prepared only with $Al(NO_3)_3$ under the same condition for SP-60. As represented by a grey curve in Figure 2a, this Al precipitate shows the reflections corresponding to the reference PDF #21-1307 for boehmite. Therefore, the $Al^{3+}$ species is likely to be precipitated as AlO(OH) in SP-derived precursors, which is supported by a dehydroxylation event of SP-60 and SP-80 observed in 300–800 K (Figure 2b). Consequently, all XRD findings explain the deposition of AlO(OH) onto the outer surface of crystalline zM material by sequential precipitation.

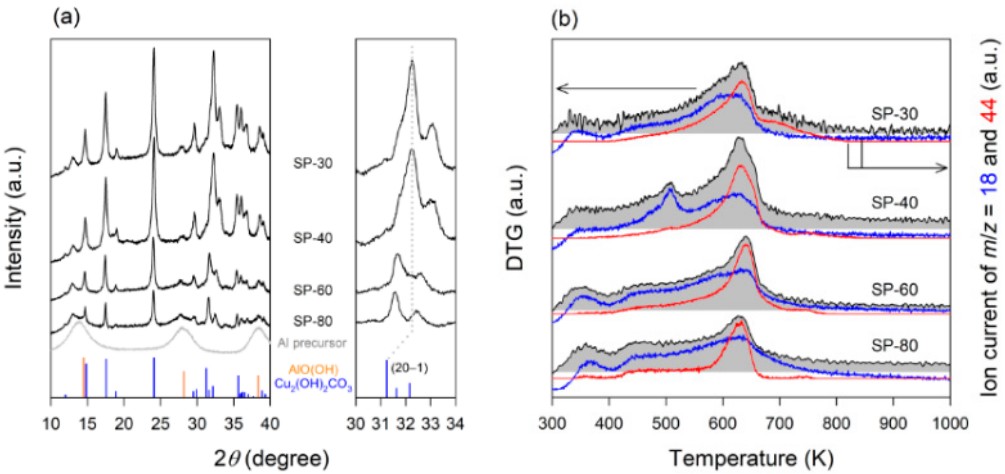

**Figure 2.** (**a**) XRD patterns of SP-derived precursors and pure Al precursor with the reference of malachite ($Cu_2(OH)_2CO_3$, blue bar) and boehmite ($AlO(OH)$, orange bar), and (**b**) DTG curves and MS traces for $H_2O$ ($m/z = 18$, blue) and $CO_2$ ($m/z = 44$, red) evolved during TG experiments.

The catalyst acidity is a crucial property in the direct conversion of $CO_2$ to DME because it largely affects the rate of methanol dehydration to DME in the presence of water produced by $CO_2$ hydrogenation to methanol and rWGS reaction. Thus, we conducted ammonia temperature-programmed desorption (TPD) experiments to determine the acidity of $Al_2O_3/Cu/ZnO$ catalysts prepared by calcination at 673 K for 3 h and subsequent $H_2$ reduction at 573 K for 5 h. Figure 3 presents TPD profiles represented by the mass fragment of $m/z = 16$. Note that $NH_3$ and $H_2O$ emissions was overlapped that was evidenced by the TCD signal and mass fragments of $m/z = 17$ and 18 (not shown for brevity). The area of the desorption peak increased with Al composition in the catalyst. When normalized by the peak area measured from SP-30, the catalyst acidity ($A_{NH3\text{-}TPD}$ in Table 1) decreased in the following order: SP-80 (1.63) > SP-60 (1.45) > SP-40 (1.37) > SP-30 (1.00). Therefore, SP-80 is expected to exhibit the highest dehydration rate among the SP-derived catalysts. When a commercial $\gamma$-$Al_2O_3$ sample (Strem Chemicals) was measured, the $A_{NH3\text{-}TPD}$ value was estimated to be 1.39, implying that SP-60 and SP-80 are more acidic than $\gamma$-$Al_2O_3$.

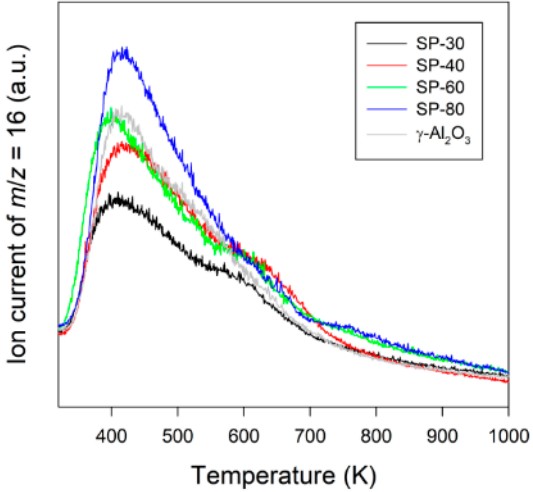

**Figure 3.** $NH_3$-TPD profiles (represented by the mass fragment of $m/z = 16$) of SP-derived catalysts and $\gamma$-$Al_2O_3$ sample.

**Table 1.** Characteristics of the prepared SP-derived samples.

| | Calcined Samples | | | | Reduced Samples | | |
|---|---|---|---|---|---|---|---|
| | Atomic percentage (%) [1] | | | $S_{BET}$ (m$^2$ g$^{-1}$) | $S_{Cu}$ (m$^2$ g$^{-1}$) | $D_{Cu}$ (%) [2] | $A_{NH3\text{-}TPD}$ [3] |
| | Cu | Zn | Al | | | | |
| SP-30 | 48.4 | 21.0 | 30.5 | 131 | 21.9 ± 2.8 | 7.9 ± 1.0 | 1.00 |
| SP-40 | 46.4 | 19.1 | 34.4 | 127 | 21.4 ± 2.9 | 7.9 ± 1.1 | 1.37 |
| SP-60 | 29.4 | 12.4 | 58.2 | 248 | 12.5 ± 2.0 | 6.6 ± 1.0 | 1.45 |
| SP-80 | 14.2 | 6.2 | 79.7 | 309 | 7.5 ± 1.1 | 7.4 ± 1.1 | 1.63 |

[1] The values of SP-30 and SP-40 are taken from our previous report [5]; [2] Cu dispersion ($D_{Cu}$) = (surface Cu atoms calculated by $N_2O$-RFC results)/(bulk Cu atoms measured by ICP-AES) × 100; [3] Normalized catalyst acidity calculated from $NH_3$-TPD profiles.

## 2.2. Direct Conversion of $CO_2$ to DME over $Al_2O_3$/Cu/ZnO Catalysts

For the direct conversion of $CO_2$ into DME, we first investigated the effect of reaction temperature on the catalytic activity of SP-60. As the temperature increased from 503 to 573 K, $CO_2$ conversion and DME selectivity were improved from 15.8% to 24.8% and from 1.6 to 53.4%, respectively (Table 2). CO yield also increased from 11.2% to ca. 15.0% due to the endothermic nature of rWGS reaction. Another finding was the pronounced formation of methane above 573 K: $CH_4$ was not detected at 503 and 523 K, but its CO-free selectivity increased from 1.8% to 10.2% with the temperature from 548 to 598 K. This is caused by the fact that methanation reaction is favorable at higher temperatures.

Thus, the prepared $Al_2O_3$/Cu/ZnO catalysts were tested at 573 K and 50 bar. Significant changes were observed in DME and methanol selectivities. SP-30 and SP-40 produced methanol as the major product, but SP-60 and SP-80 showed the enhanced DME selectivity of 53.4% and 75.1%, respectively (Table 3). This is consistent with the acidity trend of SP-derived catalysts. However, $CO_2$ conversion was similar at ca. 25.0% for all tested catalysts though DME selectivity was changed, indicating that methanol synthesis proceeds at a higher rate than methanol dehydration. Nevertheless, it is worth noting here that DME selectivity over SP-80 is higher than that over hybrid Cu-ZnO-$Al_2O_3$/HZSM-5 (65%) [12], though the reaction condition is a little different. Additionally, we prepared a physical mixture of Cu/ZnO and γ-$Al_2O_3$ (denoted as PM-80) to have the similar Al composition to SP-80. This mixture exhibited the lower $CO_2$ conversion (21.3%) and the similar DME selectivity (75.9%), compared to SP-80. $CH_4$ selectivity (8.9%) was higher as well. This result suggests that the SP-derived catalysts of higher Al compositions can transform $CO_2$ into DME efficiently.

**Table 2.** Activity results in the conversion of $CO_2$ into DME over SP-60 at different temperatures [1].

| T (K) | $X_{CO2}$ (%) | $Y_{CO}$ (%) | CO-Free Selectivity (%) | | |
|---|---|---|---|---|---|
| | | | $CH_4$ | DME | $CH_3OH$ |
| 503 | 15.8 | 11.2 | 0.0 | 1.6 | 98.4 |
| 523 | 20.4 | 13.6 | 0.0 | 3.7 | 96.3 |
| 548 | 23.9 | 15.2 | 1.8 | 17.8 | 80.4 |
| 573 | 24.8 | 14.8 | 5.4 | 53.4 | 41.2 |
| 598 | 25.4 | 15.6 | 10.2 | 50.6 | 39.2 |

[1] Reaction condition: 50 bar and GHSV = 1450 L kg$_{cat}^{-1}$ h$^{-1}$ (0.6 g catalyst).

**Table 3.** Activity results in the conversion of $CO_2$ into DME over SP-derived catalysts [1].

| Catalyst | $X_{CO2}$ (%) | $Y_{CO}$ (%) | CO-Free Selectivity (%) | | |
|---|---|---|---|---|---|
| | | | $CH_4$ | DME | $CH_3OH$ |
| SP-30 | 25.2 | 18.5 | 4.2 | 16.9 | 78.9 |
| SP-40 | 25.0 | 19.0 | 5.2 | 36.9 | 57.9 |
| SP-60 | 24.8 | 14.8 | 5.4 | 53.4 | 41.2 |
| SP-80 | 25.2 | 12.6 | 0.5 | 75.1 | 24.4 |
| PM-80 [2] | 21.3 | 10.1 | 8.9 | 75.9 | 15.2 |

[1] Reaction condition: 50 bar, 573 K and GHSV = 1450 L kg$_{cat}^{-1}$ h$^{-1}$ (0.6 g catalyst). [2] PM-80: a physical mixture of Cu/ZnO and commercial γ-$Al_2O_3$ in which Al ratio is 80%.

The activity results are supported by the numbers of surface Cu site and acid site. SP-60 and SP-80 showed the $S_{Cu}$ values of 12.5 and 7.5 m$^2$ g$^{-1}$, respectively, that were smaller than those of SP-30 and SP-40 (Table 1). This is simply due to the lower Cu loadings in the former two catalysts because Cu dispersion ($D_{Cu}$) was similar in the range 6.6%–7.9% for all catalyst samples. The distribution of Cu site over SP-60 and SP-80 was confirmed by HRTEM-EDS and SEM-EDS measurements. Figures 4 and 5 display that Cu is distributed well over the catalyst surface, indicating that the first step of methanol synthesis takes place at a similar extent although the total Cu amount is reduced with Al composition increasing. Moreover, Al-rich particles were observed in the vicinity of Cu,Zn-rich ones, which is more pronounced over SP-80. These separate Al-rich particles are believed to be responsible for the step of methanol dehydration to DME since DME selectivity was higher as the Al composition increased to 80%. Furthermore, DME selectivity varied in a similar manner with the number of acid site ($A_{NH3-TPD}$) as Al composition increased. Therefore, the highest DME selectivity was achieved over SP-80 of the most acidic nature.

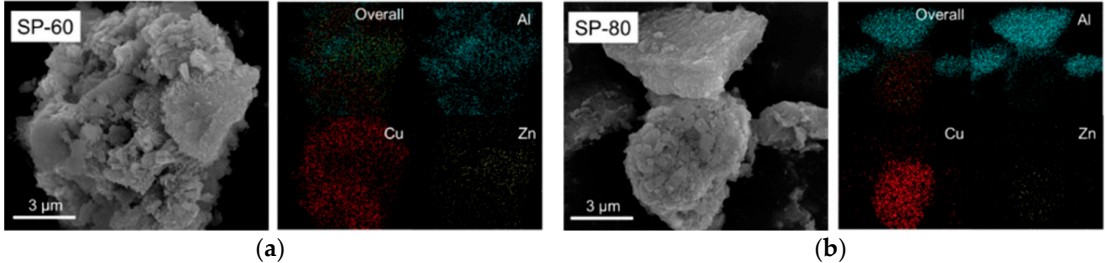

**Figure 4.** SEM and EDS mapping images of the calcined SP-60 (**a**) and SP-80 (**b**) samples.

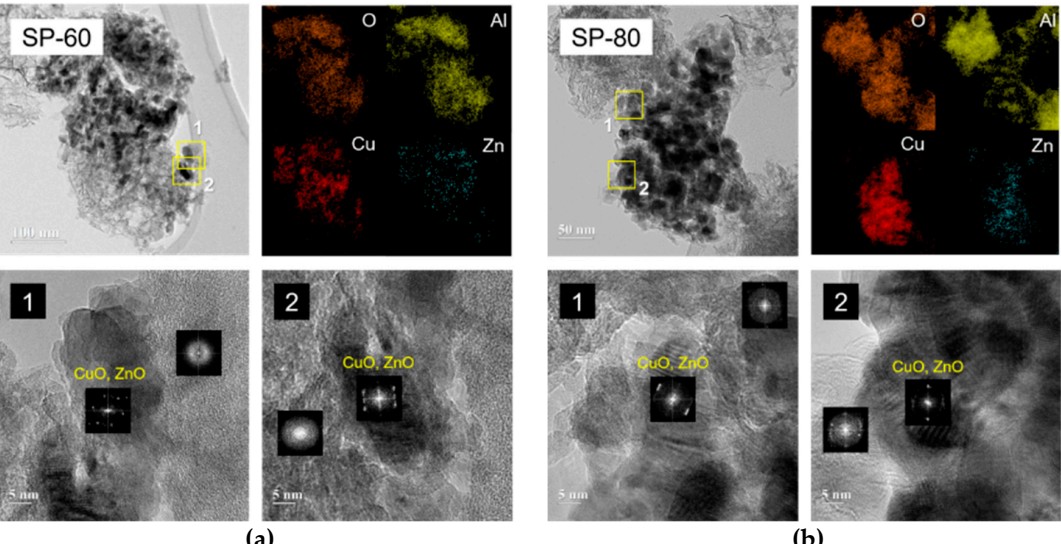

**Figure 5.** HRTEM and EDS mapping images of the calcined SP-60 (**a**) and SP-80 (**b**) samples. For each sample two different parts are magnified for taking electron diffraction patterns of local areas (inset).

### 2.3. Attempts to Improve the Selectivity to DME

To produce DME more selectively over the catalyst SP-60, we varied the reaction condition for the activity results presented in Table 2. First, the increase in the reaction pressure from 50 to 65 bar with the other variables unchanged resulted in the small increase in $CO_2$ conversion and DME selectivity by 2.1 and 8.9%, respectively (entry 1 of Table 4). Another attempt was made to lower the GHSV from 1450 to 335 L kg$_{cat}$$^{-1}$ h$^{-1}$ that is equivalent to the 4.3-fold increase in the contact time. As a result, DME selectivity improved to 78.0% though $CO_2$ conversion was negligible changed (entry 2 of Table 4). These efforts were considered meaningful because the obtained DME selectivity is comparable to or a

little higher than those of any hybrid/bifunctional catalysts reported in literature (cf. Table 10 in the review article of Saravanan et al. [3]).

Since DME selectivity over SP-60 did not exceed 80%, we tried using a dual-bed catalyst system composed of SP-60 (0.6 g) in the first layer and $\gamma$-$Al_2O_3$ (2.0 g) in the second bed, resulting in the GHSV of 335 L $kg_{cat}^{-1}$ $h^{-1}$ that is the same as the value used for the previous case. As listed in entry 3 of Table 4, DME selectivity increased to 84.1% because methanol dehydration to DME took place additionally over acid sites of $\gamma$-$Al_2O_3$. Nevertheless, CO-free methanol selectivity was measured to remain 14.3% along with no improvement of $CO_2$ conversion. This is possibly explained by the reaction pathway from $CO_2$ to DME: water, which is produced via $CO_2$ hydrogenation to methanol and rWGS reaction, limits the dehydration of methanol to DME.

**Table 4.** Activity results in $CO_2$ conversion to DME at 573 K under different reaction conditions.

| Entry | Catalyst | Pressure (Bar) | GHSV (L $kg_{cat}^{-1}$ $h^{-1}$) | $X_{CO2}$ (%) | $Y_{CO}$ (%) | CO-Free Selectivity (%) | | |
|---|---|---|---|---|---|---|---|---|
| | | | | | | $CH_4$ | DME | $CH_3OH$ |
| 1 | SP-60 (0.6 g) | 65 | 1450 | 26.9 | 15.0 | 10.9 | 62.3 | 26.8 |
| 2 | SP-60 (2.6 g) | 50 | 335 | 24.7 | 5.6 | 4.4 | 78.0 | 17.6 |
| 3 | SP-60 + $\gamma$-$Al_2O_3$ [1] | 50 | 335 | 24.6 | 12.0 | 1.6 | 84.1 | 14.3 |

[1] A dual-bed catalyst system consisting of 0.6 g SP-60 in the first layer and 2.0 g $\gamma$-$Al_2O_3$ in the second layer.

To circumvent this limitation, we endeavored to implement another series reaction, that is, DME carbonylation to methyl acetate (MA) over a suitable zeolite catalyst [18,19]. This was expected to be greatly probable in our work because CO is provided by rWGS reaction and react with DME. Prior to activity test, the effects of reaction temperature and pressure on MA yield were investigated using the ASPEN Plus software (V8.8®, Aspen Technology, Inc. Bedford, MA, USA). The simulation was based on a couple of equilibrium reactors: the first reactor includes $CO_2$ hydrogenation to methanol, rWGS reaction and methanol dehydration to DME that are the representative reactions over SP-derived catalyst, and the second reactor includes methanol dehydration to DME and DME carbonylation to MA that can take place over a zeolite catalyst. The calculation results revealed that a feasible reaction temperature and pressure exist in the range 573–598 K and 50–65 bar, respectively (Figure 6). Although different from the condition for DME carbonylation reported in literature (usually below 450 K and 20 bar [20,21]), the finding seemed to be compatible with our test condition.

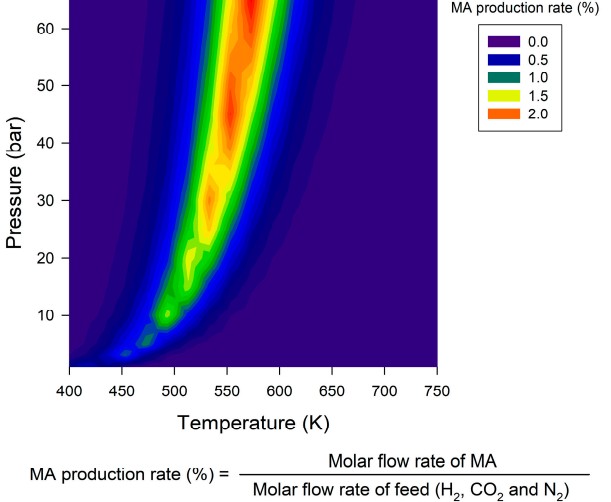

$$MA \text{ production rate } (\%) = \frac{\text{Molar flow rate of MA}}{\text{Molar flow rate of feed } (H_2, CO_2 \text{ and } N_2)}$$

**Figure 6.** Production rate of methyl acetate (MA) relative to the molar flow rate of feed ($H_2$/$CO_2$/$N_2$ = 72/24/4) as a function of the reaction temperature and pressure (calculated by ASPEN Plus).

The activity test was performed in a dual-bed catalyst system using the SP-derived catalyst and zeolite ferrierite (FER@FER) recently developed by a seed method [22], where the latter catalyst of 2.0 g was placed right after the bed of 0.6 g SP-60 or SP-80 (Table 5). The dual-bed of SP-60 and FER@FER at 573 K and 50 bar showed the increase in DME selectivity (82.2%) and the decrease in methanol selectivity (10.5%) compared to only SP-60 under the same reaction condition. Moreover, MA was also formed with the selectivity of 3.6%. When the reactor was pressurized to 65 bar, $CO_2$ conversion and DME selectivity increased to 28.8% and 90.4%, respectively, while MA selectivity decreased to 1.9% because of less formation of CO. In the dual-bed of SP-80 and FER@FER, slightly higher DME and MA selectivities were obtained owing to more acid sites of SP-80. Remarkably, the measured DME selectivities of *ca.* 90% is close to that obtained under 360 bars with a physical mixture of $Cu/ZnO/Al_2O_3$ and HZSM-5 [23]. Figure 7 displays the catalytic performance of SP-60/FER@FER dual with time on stream. During the initial run of 5 h, a large amount of methanol was shifted to DME and then DME conversion to MA happened. Afterwards, the steady-state operation was noticed. After being pressurized from 50 to 65 bar, DME selectivity increased to 90.4% and methanol selectivity vice versa (ca. 5%). Consequently, DME selectivity of over 90% was achieved by this catalyst configuration.

**Table 5.** Activity results obtained through a dual-bed catalyst system [1].

| Catalyst | P (Bar) | $X_{CO2}$ (%) | $Y_{CO}$ (%) | CO-Free Selectivity (%) | | | |
|---|---|---|---|---|---|---|---|
| | | | | $CH_4$ | DME | $CH_3OH$ | MA |
| SP-60 and | 50 | 24.8 | 11.5 | 3.7 | 82.2 | 10.5 | 3.6 |
| ferrierite | 65 | 28.8 | 9.8 | 2.8 | 90.4 | 4.9 | 1.9 |
| SP-80 and | 50 | 26.6 | 12.5 | 5.4 | 83.0 | 7.3 | 4.3 |
| ferrierite | 65 | 27.9 | 10.7 | 2.3 | 90.9 | 4.6 | 2.2 |

[1] Reaction condition: 573 K and GHSV = 335 L $kg_{cat}^{-1}$ $h^{-1}$ (0.6 g SP-60 or SP-80 and 2.0 g ferrierite).

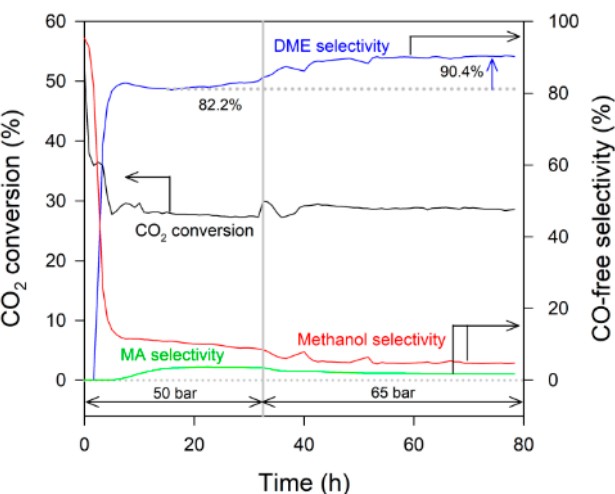

**Figure 7.** $CO_2$ conversion (**black**) and CO-free selectivities to methanol (**red**), DME (**blue**) and MA (**green**) in a dual-bed catalyst system (0.6 g SP-60 and 2.0 g FER@FER) at 573 K and GHSV of 335 L $kg_{cat}^{-1}$ $h^{-1}$. The reactor pressure was initially set at 50 bar and then increased to 65 bar at the time on stream of 30 h (marked by a vertical dotted line).

A schematic of the demonstrated dual-bed catalyst system is depicted in Figure 8. In the first bed, the SP-derived catalyst enables direct $CO_2$ conversion to DME. However, there exists the limitation associated with water which is formed in the course of $CO_2$ transformation into both methanol and CO (via rWGS). This lowers the rate of methanol dehydration over $Al_2O_3/Cu/ZnO$. The next bed is packed with FER@FER converting a fraction of DME to MA via carbonylation using the produced CO. This second catalyst also facilitates the overall reaction starting from $CO_2$ to DME at its acid sites,

accordingly DME selectivity increasing to 90%. Therefore, this dual-bed system contributes largely to selective DME production and simultaneously converts DME, even if small, into MA that can be further changed to ethanol by hydrogenolysis or acetic acid via hydrolysis.

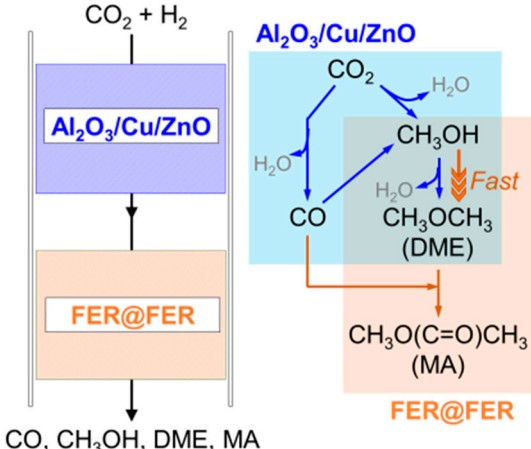

**Figure 8.** Scheme of dual-bed catalyst system: SP-derived catalyst for $CO_2$ hydrogenation to methanol, methanol dehydration to DME and rWGS reaction (**blue-shaded part**) coupled with FER@FER for DME formation and DME carbonylation to MA (**orange-shaded part**).

## 3. Materials and Methods

### 3.1. Catalyst Preparation

A series of $Al_2O_3$/Cu/ZnO catalysts were prepared by the similar preparation method that was previously reported as sequential precipitation [6]. In order to make complete precipitation, the concentration of precipitation agent was controlled from 0.1 to 0.16 M. Based on the calculation of the $NaHCO_3$ amount, the 4200 mL basic solution in a 5-L precipitation reactor was preheated to 343 K. The aqueous 1.2 M $Cu^{2+}$ and $Zn^{2+}$ solution ($Cu^{2+}/Zn^{2+}$ = 70:30) was injected dropwise (14 $cm^3$ $min^{-1}$) under vigorous stirring. After first ageing at the same temperature for 1 h, the $Al^{3+}$ solution was added at the same injection rate and kept stirring for 30 min for further ageing. The $Al^{3+}$ amount and water volume of both metal ion solutions were calculated based on the desired nominal Al content ($x$ = [Al]/{[Cu] + [Zn] + [Al]} × 100%). The final aged precipitate was thoroughly washed with deionized water for four times and filter cake was dried in a convection oven at 378 K overnight. The prepared precursor was labelled as SP-$x$Al, where $x$ is the nominal Al content ([Al]/{[Cu] + [Zn] + [Al]} × 100%). All the prepared precursor samples were crushed and sieved to the size smaller than 200 μm, followed by calcination in a muffle furnace at 673 K (5 K $min^{-1}$) for 3 h. On the other hand, the zeolite ferrierite was prepared using a recipe reported by Bae et. al. [22].

### 3.2. Characterization

X-ray diffraction (XRD) analysis was conducted in a miniFlex600 (Rigaku) using a Cu K$\alpha$ radiation (40 kV and 15 mA). Thermogravimetric (TG) profiles were obtained in a NETZSCH TG209F1 as the sample was heated to 1073 K at a rate of 10 K $min^{-1}$ in an air flow (100 $cm^3$ $min^{-1}$). Scanning electron microscopy (SEM) images were taken in a FEI (Hillsboro, OR, USA) Nova NanoSEM 450 microscope after the sample was coated by Pt. For high resolution transmission electron microscopy (HR-TEM) analysis coupled with electron diffraction, a JEOL (Tokyo, Japan) JEM 2100F microscope was used with the Gatan DigitalMicrograph imaging filter. The Brunauer-Emmett-Teller (BET) surface area of a sample (0.15 g) was measured in a Micromeritics ASAP 2020 after pretreatment at 373 K for 1 h under vacuum. The metal compositions of oxide samples were measured by inductively coupled plasma atomic emission spectroscopy (ICP-AES) using an Optima 8300 (Perkin-Elmer, Waltham, MA, USA). The temperature-programmed reduction (TPR) experiment was conducted in a Micromeritics

AutoChem 2900. As the sample (50 mg) was heated at a rate of 2.5 K min$^{-1}$ using 10% $H_2$ in Ar (50 cm$^3$ min$^{-1}$), the effluent gas was measured by a quadruple mass spectroscope (Balzers Prisma QMS 200, Pfeiffer Vacuum, Aßlar).

For $N_2O$ reactive frontal chromatography ($N_2O$-RFC) experiment to measure copper surface area, the sample (0.1 g) was reduced at 573 K for 1 h (5 K min$^{-1}$) using 10% $H_2$ in Ar (30 cm$^3$ min$^{-1}$) in a BELCAT-B (BEL Japan, Inc.). After cooling to 313 K in He, 1% $N_2O$ in He (5 cm$^3$ min$^{-1}$) was introduced and product gas ($N_2$, *m/z* = 28) was measured by a BEL-Mass (BEL Japan, Inc.). It was assumed that the reaction stoichiometry between copper and oxygen is two (Cu/O = 2/1) and the copper surface density is $1.46 \times 10^{19}$ Cu atom m$^{-2}$. Although $N_2O$-RFC result cannot be directly related to only the exposed Cu surface [24,25], the measured copper surface area still remains the best indicator for the catalytic performance of Cu/ZnO-based catalysts.

For ammonia temperature-programmed desorption ($NH_3$-TPD) experiment to measure the amount of acidity, the calcined sample (0.1 g) was reduced at 573 K for 1 h (5 K min$^{-1}$) using 10% $H_2$ in Ar (30 cm$^3$ min$^{-1}$) in a BELCAT-B (BEL Japan, Inc.). After cooling to 323 K in He, 5% $NH_3$ in He (30 cm$^3$ min$^{-1}$) was introduced for ammonia adsorption, followed by He purge at 323 K for 40 min. As the sample was heated at a rate of 10 K min$^{-1}$ in He (30 cm$^3$ min$^{-1}$), the effluent gas was measured by a BELCAT-B and a BEL-Mass (BEL Japan, Inc.).

*3.3. Activity Test*

For $CO_2$ to DME reaction, the mixed oxide (0.6 g) was reduced at 573 K for 5 h (5 K min$^{-1}$) using 20% $H_2$ in $N_2$ (100 cm$^3$ min$^{-1}$) in a stainless-steel reactor. After cooling to a desired temperature, the reaction gas ($H_2$/$CO_2$/$N_2$ = 72/24/balance, 14.5 cm$^3$ min$^{-1}$, GHSV = 1450 L kg$_{cat}$$^{-1}$ h$^{-1}$) was fed into the reactor and pressurized to a desired value. When zeolite ferrierite was used for DME conversion, the catalyst reduction was done under the same condition as stated above and the volumetric flow rate of reaction gas was also identical to 14.5 cm$^3$ min$^{-1}$.

## 4. Conclusions

We have developed $Al_2O_3$/Cu/ZnO catalysts of 60%–80% Al compositions for $CO_2$ hydrogenation to dimethyl ether. All the prepared precursors were composed of crystalline zincian malachite and amorphous AlO(OH), thus resulting in the similar Cu dispersion while Al composition was higher. Moreover, Al-rich $Al_2O_3$/Cu/ZnO catalysts retained acid sites responsible for methanol dehydration to DME. Owing to this bifunctional character, the catalysts SP-60 and SP-80 could yield DME as the major product. Additionally, each of these two catalysts was coupled with FER@FER for dual-bed catalyst system which demonstrated selective DME production (e.g., CO-free selectivity > 90%). Consequently, the suggested process scheme would be viable due to the bifunctionality of the SP-derived catalysts and possibly contributes to the commercial production of DME from $CO_2$.

**Author Contributions:** Conceptualization, C.J., J.W.B. and Y.-W.S.; methodology, C.J. and J.K.; software, S.L.; validation, C.J. and J.K.; formal analysis, C.J. and J.K.; investigation, C.J., J.K., J.-H.K.; data curation, C.J. and J.K.; writing—original draft preparation, C.J.; writing—review and editing, C.J. and Y.-W.S.; visualization, C.J.; supervision, Y.-W.S.; project administration, Y.-W.S.; funding acquisition, Y.-W.S.

**Funding:** This research was supported by C1 Gas Refinery Program (Project No. 2018M3D3A1A01018010) through the National Research Foundation (NRF) funded by the Ministry of Science and ICT, Republic of Korea, as well as by Basic Science Research Program (Project No. 2016R1A6A1A03013422) through the NRF funded by the Ministry of Education, Republic of Korea.

**Conflicts of Interest:** The authors declare no conflict of interest.

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
