# Peer review of "Direct Conversion of CO2 into Dimethyl Ether over Al2O3/Cu/ZnO Catalysts Prepared by Sequential Precipitation"

_catalysts, doi:10.3390/catal9060524_

Round 1

Reviewer 1 Report

New bifunctional Al2O3/Cu/ZnO catalysts for the direct conversion of CO2 into dimethyl ether (DME) have been developed, leading to acceptable CO2 conversions and selectivities for this relevant transformation. The main novelty of this work is the use of a sequential precipitation strategy for the preparation of the catalyst that allows to generate Al-enriched systems. Overall it is an interesting manuscript that deserves publication in Catalysts after addressing the following minor points:

- The review article by Catizzone and co-workers on CO2 to DME conversion (Molecules 2018, 23, 31) should be included in the references list.

- The authors should also compare and discuss the conversions and selectivities achieved with those of related catalytic systems already published in the literature.

- Given that IUPAC and NIST agency discourage the use of normality (N) as unit of concentration, it would be appreciated the authors employ molarity concentration all along the manuscript.

Author Response

The review article by Catizzone and co-workers on CO2 to DME conversion (Molecules 2018, 23, 31) should be included in the references list.

Response: The named review article has been included in the Reference.

The authors should also compare and discuss the conversions and selectivities achieved with those of related catalytic systems already published in the literature.

Response: According to the reviewer’s suggestion, we have compared our reported activity results with those of related catalyst systems published in literature. New discussions have been made at three different places, as follows:

Line 160–162 in the revised manuscript:

Nevertheless, it is worth noting here that DME selectivity over SP-80 is higher than that over hybrid Cu-ZnO-Al2O3/HZSM-5 (65%) [12], though the reaction condition is a little different.  

Line 197–199 in the revised manuscript:

These efforts were considered meaningful because the obtained DME selectivity is comparable to or a little higher than those of any hybrid/bifunctional catalysts reported in literature (cf. Table 10 in the review article of Saravanan et al. [3]).

Line 235–237 in the revised manuscript:

Remarkably, the measured DME selectivities of ca. 90% is close to that obtained under 360 bar with a physical mixture of Cu/ZnO/Al2O3 and HZSM-5 [23].

Given that IUPAC and NIST agency discourage the use of normality (N) as unit of concentration, it would be appreciated the authors employ molarity concentration all along the manuscript.

Response: The normality concentration of NaHCO3 solution is the same as the molarity concentration, as the reviewer knows well. According to the suggestion, we have changed the unit from the normality to molarity concentration at appropriate places.

Reviewer 2 Report

This work presents a series of study on Al/Cu/Zn based catalysts for CO2 reduction to DME by varying Al content, test conditions, and catalyst system design to achieve high DME selectivity and CO2 conversion. While this paper contains a lot of work, a few fundamental questions need to be addressed before it can be published on a catalyst-focused journal.

The manuscript needs to clarify:

(1)    What is the role of Al in the reaction and why the authors to choose varying the Al content?

(2)    Introduction line 56-59, how does the two precipitation methods affect the catalyst structure and performance. This needs to be clarified with the mechanism of CO2 reduction and selectivity of different product. A lot of references are needed here.

(3)    Again, introduction line 63-67. References are needed here to explain the DME selectivity. By the way, the two reaction equations are missing arrows.

(4)    Line 68-79, consider reorganize to make the motivation/objectives straight to the point.

(5)    What is the rationale of testing the acidity of the catalyst? This has to be correlated to the reaction mechanisms (in the introduction section) without which it is impossible to discuss.

(6)    Does the optimum test condition vary with different catalyst compositions (i.e. different SP- values)?

(7)    Line 190-191, what are the reaction pathways?

(8)    Finally, a quite important one, while a DME selectivity of >90% is achieved with dual-bed catalysts design, the CO2 conversion % does not improve much (from ~24% to ~28%). Any comments? Personally, the CO2 conversion % may be more important than product selectivity.

Author Response

For your convenience, all comments raised by the reviewers are reproduced in blue italic while the revision made is highlighted in red.

What is the role of Al in the reaction and why the authors to choose varying the Al content?

Response: As the reviewer may know well, the conversion of CO2 to DME requires two catalytic functions such as metallic and acidic sites responsible for CO2 hydrogenation to methanol and subsequent dehydration to DME, respectively. Therefore, Al2O3 acts as acidic site for the latter reaction.

Our approach to increase Al composition in the SP-derived Al2O3/Cu/ZnO catalysts has a very simple reason. That is, SP-30 and SP-40 reported in our previous work did not work well in direct conversion of CO2 to DME, as we mentioned in the Introduction. Thus, SP-60 and SP-80 with higher Al compositions are newly prepared in this work. These catalysts contain more acidic sites, which is demonstrated by ammonia TPD experiment, and thus show higher DME selectivity values.

Introduction line 56-59, how does the two precipitation methods affect the catalyst structure and performance. This needs to be clarified with the mechanism of CO2 reduction and selectivity of different product. A lot of references are needed here.

Response: We appreciate the reviewer’s comment. In general, homogeneous co-precipitation leads to formation of well-mixed Cu, ZnO, and Al2O3 that favors methanol synthesis by CO and/or CO2 hydrogenation, where Al2O3 is a structural promoter for Cu particles necessary for methanol synthesis. In contrast, sequential precipitation (SP) we developed enabled Al2O3 to exist at the external surface of Cu/ZnO core. Due to this structural difference, the SP-derived Al2O3/Cu/ZnO catalysts could efficiently convert carbon monoxide into methanol and then into DME, compared to their co-precipitated counterparts, because CO hydrogenation to methanol took place over a greater number of Cu surface atoms and subsequent dehydration of methanol to DME was facilitated by Al2O3 acid sites existing at the external particle surface. Therefore, Al2O3 in the SP-derived Al2O3/Cu/ZnO catalysts acts as acid site for the dehydration reaction.

Based on the above explanation, the sentences in line 56–59 have been revised as follows:

Our strategy is so-called sequential precipitation (SP), different from homogeneous co-precipitation that produces a well-mixed phase of Cu, ZnO, and Al2O3 favoring the formation of methanol by CO and/or CO2 hydrogenation [7,8]. The SP-derived Al2O3/Cu/ZnO catalysts could efficiently convert carbon monoxide into methanol and then into DME, compared to their co-precipitated counterparts, because CO hydrogenation to methanol took place over a greater number of accessible Cu surface atoms and subsequent dehydration of methanol to DME was facilitated by Al2O3 acid sites existing at the external particle surface [6]. In other words, Al2O3 in conventional co-precipitated Cu/ZnO/Al2O3 is a structural promoter for Cu particles necessary for methanol synthesis [9,10], whereas Al2O3 in the SP-derived Al2O3/Cu/ZnO catalysts acts as acid site for the dehydration reaction [11,12].

Again, introduction line 63-67. References are needed here to explain the DME selectivity. By the way, the two reaction equations are missing arrows.

Response: Arrows have been inserted in the two reaction equations. Also, the following statement has been newly added with references.

This explains that DME selectivity from CO2 is thermodynamically less than that from CO [3,4,13].

Line 68-79, consider reorganize to make the motivation/objectives straight to the point.

Response: According to the reviewer’s comment, we have reorganized the last part of Introduction section, as follows:

Since such a negative effect of CO2 on DME selectivity is considered to be compromised by a higher acidity of SP-derived Al2O3/Cu/ZnO, the composition of Al needs to be increased up to 80% in the total metal elements. Thus, the catalysts with 60 and 80% Al were prepared by using a higher concentration of precipitation agent to induce complete precipitation of all metal ions, aside from Al2O3/Cu/ZnO with 30 and 40% Al. All of the prepared catalysts were tested in direct CO2 conversion to DME. Then, we tried to screen the reaction condition and use γ-Al2O3 in the second catalyst bed to improve DME selectivity. To overcome the observed limitation in DME selectivity caused by thermodynamic equilibrium, we considered additional reaction that can convert DME because the equilibrium would be shifted to a relatively higher level by consumption of DME. We attempted to place zeolite ferrierite in the second bed after the first Al2O3/Cu/ZnO bed, which was successful for higher DME selectivity while methyl acetate was formed by DME carbonylation using CO produced via rWGS reaction. Consequently, the catalyst systems we investigated herein would pave the way for converting CO2 into DME in a selective manner.

What is the rationale of testing the acidity of the catalyst? This has to be correlated to the reaction mechanisms (in the introduction section) without which it is impossible to discuss.

Response: As presented above, we have revised the Introduction in order to make clear the reaction mechanism involved in direct CO2 conversion to DME. Since the catalyst acidity affects the rate of methanol dehydration to DME, we should measure it in a quantitative manner. The technique we chose in this work is ammonia TPD experiment. According to the reviewer’s comment, we have newly added discussion on the correlation of catalyst acidity with the reaction mechanism, as follows:

The catalyst acidity is a crucial property in direct conversion of CO2 to DME because it largely affects the rate of methanol dehydration to DME in the presence of water produced by CO2 hydrogenation to methanol and rWGS reaction. Thus, we conducted ammonia TPD experiments to determine the acidity of Al2O3/Cu/ZnO catalysts prepared by calcination at 673 K for 3 h and subsequent H2 reduction at 573 K for 5 h.

Does the optimum test condition vary with different catalyst compositions (i.e. different SP- values)?

Response: The selectivity to DME depends upon Al composition of different SP-xAl catalysts. However, thermodynamic equilibrium cannot be altered by Al composition. This is why we investigated an optimal reaction condition using the ASPEN Plus software. Therefore, the testing condition we applied for this work does not vary with catalyst compositions.

Line 190-191, what are the reaction pathways?

Response: The reaction pathway stated in those lines includes CO2 hydrogenation to methanol, rWGS reaction, and methanol dehydration to DME. Thus, the corresponding sentence has been modified as follows:

This is possibly explained by the reaction pathway from CO2 to DME: water, which is produced via CO2 hydrogenation to methanol and rWGS reaction, limits the dehydration of methanol to DME.

Finally, a quite important one, while a DME selectivity of >90% is achieved with dual-bed catalysts design, the CO2 conversion % does not improve much (from ~24% to ~28%). Any comments? Personally, the CO2 conversion % may be more important than product selectivity.

Response: As we often mentioned in the original manuscript, the conversion of CO2 to DME has strong thermodynamic limitation due to water molecules produced by CO2 hydrogenation to methanol, compared to CO hydrogenation to methanol. One of the authors already reported that the thermodynamic CO2 conversion was 2- to 3-fold lower that CO conversion at 573 K [2]. This is why we focused on selective formation of DME while keeping or slightly increasing CO2 conversion, although we all know well about the importance of CO2 conversion value. We hope that the response to this comment is satisfactory.

Round 2

Reviewer 2 Report

I thank the authors for addressing my comments. I'd like that the response to the first comment (the role of acidic sites in DME selectivity) be added to the introduction so that there is more fundamental information available to the reader. Other than that, the paper is good for publication and there is no need to have another round of review. 

Author Response

I thank the authors for addressing my comments. I'd like that the response to the first comment (the role of acidic sites in DME selectivity) be added to the introduction so that there is more fundamental information available to the reader. Other than that, the paper is good for publication and there is no need to have another round of review.

Response: We appreciate the helpful comment of the reviewer. The response to the first comment was not added into the revised manuscript since we thought that the similar discussion was already made at several places of Introduction (highlighted by underlines), as appending below:

Line 46–48: the most intensively studied reaction pathway is direct conversion of CO2 into dimethyl ether (DME) because methanol intermediate is simply changed into DME via intermolecular dehydration over acid catalysts,

Line 59–63: The SP-derived Al2O3/Cu/ZnO catalysts could efficiently convert carbon monoxide into methanol and then into DME, compared to their co-precipitated counterparts, because CO hydrogenation to methanol took place over a greater number of accessible Cu surface atoms and subsequent dehydration of methanol to DME was facilitated by Al2O3 acid sites existing at the external particle surface [6].

Line 77–79: Since such a negative effect of CO2 on DME selectivity is considered to be compromised by a higher acidity of SP-derived Al2O3/Cu/ZnO, the composition of Al needs to be increased up to 80% in the total metal elements.

   Nevertheless, we have added one more discussion into an appropriate place in order to reflect the reviewer’s comment, as follows:

The usual way to achieve this direct conversion is mixing of Cu/ZnO-based methanol synthesis catalyst with an acid catalyst such as γ-Al2O3 and zeolite by physical mixing or precipitation, where the acid catalyst is responsible for methanol dehydration to DME and therefore increases DME selectivity [2–4].